# On the Merits of Targeted and Individualized Physical Exercise in Persons with Diabetic Foot Disease—From Controversies to Consensus

**DOI:** 10.3390/biomedicines13071752

**Published:** 2025-07-17

**Authors:** Edyta Sutkowska, Anna Korzon-Burakowska, Karolina Biernat

**Affiliations:** 1Department of Clinical Physiotherapy and Rehabilitation, Wroclaw Medical University, Borowska Str. 213, 50-556 Wroclaw, Poland; karolina.biernat@umw.edu.pl; 2Department of Hypertension and Diabetology, Medical University of Gdansk, 80-210 Gdańsk, Poland; anna.korzon-burakowska@gumed.edu.pl

**Keywords:** diabetic foot disease, exercise, physical activity, diabetes, foot offloading

## Abstract

Exercise is a cornerstone of diabetes management, but the onset of diabetic foot disease (DFD) can significantly limit its implementation. Meanwhile, physical activity (PA) has been shown to reduce the risk of developing DFD through various mechanisms, and emerging evidence also supports the role of exercise in managing the active phase of the condition. Appropriately tailored PA offers both local and systemic benefits—even in clinical contexts where foot offloading is recommended. The research indicates that selected exercises can be safely incorporated into care plans, providing therapeutic effects without compromising wound healing. Drawing from current knowledge based on basic science, clinical research, and relatively general recommendations, this article summarizes the local and systemic effects of properly selected exercises in patients with DFD. It explains the underlying mechanisms and briefly discusses practical examples, integrating the most recently published findings.

## 1. Introduction

Exercise is the basis of diabetes mellitus (DM) treatment [1,2,3], as physical activity (PA) plays an important role in regulating glucose levels in the body [4,5,6,7]. Therefore undertaking activity by patients is recommended by all scientific societies dealing with the issue of carbohydrate metabolism disorders. Physical activity refers to any bodily movement produced by skeletal muscles that requires energy expenditure, whereas exercise is a structured and purposeful form of PA designed to improve physical fitness or health outcomes. Difficulties in engaging in PA among patients with DM are common, and stem from numerous issues described in observational studies. These problems arise both from the nature of the disease itself (e.g., glycemic fluctuations, risk of hypoglycemia depending on therapy, presence of complications) as well as from non-disease-related barriers (e.g., socioeconomic status, education, patient age, lack of time, etc.) [8,9,10,11].

Despite ongoing updates, the guidelines regarding recommended PA remain fairly general, and are mostly targeted at patients with DM who do not have significant complications. This is understandable, as the presence of chronic complications (including those affecting the foot) in diabetes requires a highly individualized approach to all recommendations, including those related to exercise. Nonetheless, physical activity remains essential at every stage of diabetes management, although its previous form often requires significant modification.

According to the International Working Group on the Diabetic Foot (IWGD) [12], diabetic foot disease (DFD) is defined as a condition when a patient with diabetes and neuropathy and/or arterial blood flow disorders in the lower limbs develops ulceration, infection, and/or destructive changes in the deep tissues below the ankle. Approximately 4–10% of patients with DM have active foot lesions (such as ulcers, infections, or deformities), while the lifetime risk of developing a foot ulcer reaches 15–25%, meaning that every fourth to sixth person with DM will experience this complication. In Europe, diabetes-related amputations account for 40–70% of all lower-limb amputations [13,14]. The St. Vincent Declaration, an international document developed in 1989 and signed by many European Union countries, outlined ambitious health goals to be achieved within 5–10 years of joining the initiative [15]. Among these cornerstones of health policy was the assumption that the number of lower extremity amputations caused by diabetes should be reduced by half. This particular objective proved to be one of the most difficult to achieve, due to significant regional disparities and limitations within healthcare systems. In regions where multidisciplinary diabetic foot care programs were implemented (including specialist teams, rapid intervention protocols, and patient education), significant reductions in amputation rates were observed—reaching even 50–68%. Unfortunately, on a broader population level, many regions failed to meet the target set by the declaration [16].

There is ample evidence that targeted PA reduces the risk of peripheral neuropathy (prevention stage of DFD) [17]. Adequate exercise improves sole load distribution, skin sensitivity, peripheral nerve conduction, intraepidermal nerve fiber density or joint mobility of the foot which translates into proper gait [18,19], and reduces the risk of pathologies typical of diabetes occurring in that area [20,21,22,23,24,25,26,27]. As in patients without DM, both the reduction of atherosclerosis risk factors and walking have a beneficial effect on peripheral circulation, reducing the risk of clinically significant limb ischemia leading to ulcers and amputations [28,29,30]. At the prevention stage of DFD, patients are educated about the role of exercise and its safety. The condition of the peripheral nervous system (perception of injury) and blood supply to the limbs is assessed.

Although, as mentioned, physical activity is recommended, in some cases, due to overlapping diseases, patients are advised to discontinue or limit their current activities for the duration of treatment. This may have adverse effects not only on the patients’ glycemic control (by increasing insulin resistance) and body weight but on respiratory and cardiovascular systems, and may also contribute to musculoskeletal disorders, which can translate into delayed recovery and poorer functioning after recovery [31,32,33,34,35,36,37,38]. An example of such a situation in the life of a patient with DM is the occurrence of DFD.

In the case of the presence of an unhealed ulcer, current guidelines on the treatment of diabetes and its complications provide only the recommendation to eliminate weight-bearing activity [39,40]. In the presence of Charcot deformity, the role of stretching and muscle-strengthening exercises is noted as potentially helpful [39]. In the Clinical Practice Guideline published in 2023 [41] the authors address this issue and explicitly raise the problem of “Various fitness/exercise routines with various intensities, including aerobic and resistance training” in the context of an expected outcome defined as “Improved mobility in the context of wound healing.” However, even in this document, the lack of precise recommendations regarding which exercises should be prescribed and what their potential benefits are is emphasized. Similarly, in the IWGDF guidelines [40]—the most authoritative document in this field—29 recommendations are provided for managing DFD, but none of them refer to exercise. This situation results from the limited number of RCTs (randomized controlled trials) dedicated to this topic that have included a sufficiently large number of patients in the intervention.

The aim of the article is to highlight the role that exercise, when performed with individualized precautions, can play in improving the condition of a patient with a wound and/or foot deformity due to DFD.

## 2. Basis for the Narrative Review

The foundation of our narrative review article is a comprehensive systematic review published in 2023 [42], which summarized the limited but growing body of clinical research focused on exercise interventions initiated after a diagnosis of DFD. This review highlighted the potential benefits of various exercise modalities based on the available studies. Following its publication, a new randomized clinical trial was released [43], further supporting the inclusion of exercise in the management of patients with DFD—although it has not yet influenced clinical guidelines [39,40].

In addition to the available clinical studies discussed in the mentioned systematic review and new data from 2024, we have incorporated findings from the basic research, including preclinical experiments (on animals and humans), which provide mechanistic insights into how even simple physical exercises may contribute to both local and systemic improvements in individuals with DFD.

Step by step, we aim to demonstrate that the application of exercise in patients with DFD is safe and may be recommended—even before large-scale, high-quality trials become available—provided it is applied judiciously. In the meantime, clinicians may consider relying on existing evidence from both basic science and clinical research to guide individualized therapeutic decisions.

The authors of the publication have medical (diabetology: ES, AK-B, vascular: ES diseases) and physiotherapeutic (ES, KB) backgrounds, as well as many years of experience in treating patients with DFD.

## 3. Consequences of Diabetic Foot Disease

Interventions in DFD occur at every stage of the disease and aim to prevent the onset of DFD, treat the condition itself, and prevent its consequences—particularly long-term consequences—that may lead to disability in the patient. The functional and physical consequences of DFD are presented in Table 1. This information has been compiled by the authors based on current knowledge and practical experience in managing patients with diabetic foot disease.

Diabetic foot disease substantially impairs patients’ mobility due to pain, ulcerations, infections, and the mandatory offloading of the affected limb. These limitations often lead to reduced PA, causing muscle weakness and diminished endurance, which further restricts the range of motion (ROM) and the ability to ambulate effectively. Consequently, patients with DFD face a heightened risk of falls and subsequent injuries [44,45]. Furthermore, sensory neuropathy associated with DFD disrupts proprioceptive feedback, resulting in impaired postural control and balance instability. The motor neuropathy contributes to foot deformities and altered biomechanics, exacerbating postural instability and increasing the likelihood of gait abnormalities [46,47]. At the musculoskeletal level, chronic inflammation, immobilization, neuropathic, and ischaemic changes lead to muscle atrophy and joint stiffness, reducing flexibility and strength. Additionally, bone integrity is compromised due to osteopenia or osteoporosis and structural deformities like Charcot foot, which further diminish functional capacity [48,49].

Overall, the combined effects of reduced mobility, impaired postural control, and musculoskeletal deterioration culminate in significant loss of independence. Patients with DFD often require assistance for the activities of daily living and rehabilitation interventions to improve functional outcomes and quality of life [50,51].

**Table 1 biomedicines-13-01752-t001:** Pathomechanisms and Consequences of Diabetic Foot Disease ^A^.

Stage/Event	Pathophysiological Description	Physical Consequences	Functional Consequences
1. Neuropathy (sensory, motor, autonomic)	loss of protective sensation, muscle weakness, dry/cracked skin	foot deformities, skin fissures	increased risk of injury, impaired proprioception, need for rehabilitation
2. Angiopathy (ischemia)	impaired micro- and macrovascular circulation, tissue hypoxia	cold, pale foot, diminished or absent pulses	claudication, poor wound healing, increased risk of amputation revascularization if possible, need for rehabilitation
3. Ulcer formation	due to unnoticed trauma or pressure in insensate areas	chronic wounds, often slow to heal	difficulty walking, need for footwear modification or reduced mobility, need for rehabilitation
4. Infection	rapid bacterial colonization of ulcers, potentially spreading to deep tissues	swelling, redness, abscess, osteomyelitis	hospitalization, immobilization, rehabilitation *
5. Tissue necrosis and gangrene	resulting from uncontrolled infection and/or ischemia	irreversible tissue damage, necrotic areas	surgical intervention or amputation often required, need for rehabilitation
6. Amputation (partial or complete)	removal of necrotic tissue to prevent systemic infection or preserve life/function	loss of limb or part of the limb/foot, postural imbalance	loss of independence, need for rehabilitation, psychological impact, social isolation

* In cases of sepsis, rehabilitation should be postponed or avoided until the infection is fully controlled. *^A^ Sources: Compiled by the authors based on data from* [13,40,45,46,47,50,51].

## 4. The Role of Exercises in the Therapeutic Process of Diabetic Foot Disease

### 4.1. Multimodal Treatment of DFD

In the event of a complication, such as diabetic foot, the primary goals achieved through appropriate pharmacotherapy include control of glycemia, correction of other risk factors (cholesterol level and blood pressure values), use of antiplatelet drugs, sometimes low doses of anticoagulants (applies to treatment of limb ischemia) and, in justified cases, administration of antibiotics and painkillers.

If necessary, procedures such as revascularization (in the case of vascular lesions), removal of necrotic tissue, or orthopedic interventions should be performed prior to initiating physical exercise [12,17,18,19,20,21,22,23,24,25,26,27,28,29,30]. Physical therapy could also be useful, serving as a complementary approach [52]. Regardless of the above, however, the essential recommended management of ulceration, deformation, and deep tissue inflammation in DFD is the offloading of the affected area [53,54,55]. The goal is to protect damaged tissues from further injury and to promote healing by relieving pressure during loading. Offloading often requires patient immobilization or major activity restriction [53,54], depending on the method used—such as a wheelchair, crutches, special footwear, or a total contact cast (TCC).

Thus, in the described clinical scenario, a therapeutic dilemma emerges. While local pathological changes necessitate offloading—requiring the patient to limit weight-bearing activities such as ambulation and prolonged standing [56]—these same changes, particularly inflammation, simultaneously warrant the implementation of PA, as structured exercise can exert beneficial effects on both the patient’s systemic condition and the local tissue microenvironment [28,29].

In 2023, an expert position was published [57], summarizing the recommended actions for DFD. These actions include an assessment of the usefulness of exercises that the patient may undertake during the treatment of the condition in question. While emphasizing the lack of sufficient evidence that exercise directly accelerates wound healing, the authors recommend performing some of them due to the potential benefits demonstrated both in animals [58,59] and in humans [25].

The possibility of adverse effects is an inherent aspect of any therapy. Improperly conducted exercise may exacerbate local pathological changes and increase the risk of amputation; however, well-organized medical care, adherence to the principles of comprehensive treatment, and timely response to treatment ineffectiveness minimize this risk [31].

The rather modest work on the subject of exercise by patients with diabetic foot to date is supplemented by a recent report [43], released after the publication of the above-mentioned expert study [57]. This study also supported the incorporation of exercise into the therapy of diabetic foot disease. The condition for undertaking exercises is to adhere to the principle of relieving the local change and its surrounding area.

### 4.2. The Effect of Exercise on the Local Condition in DFD

Prolonged inflammation can lead to bone resorption due to local vasodilation and increased flow in the microcirculation [60]. Long-term offloading can also provoke local osteoporosis and can generate it in other areas of the relieved limb [38]. Each of these disorders has an adverse local impact but also worsens many of the patient’s functions [33,34,35,36,37,38]. The implementation of properly selected exercises can be helpful and have measurable benefits [61,62], preventing further reduction in bone mineral density (BMD), having also a beneficial effect on glycemic control and the patient’s well-being [38].

Even local inflammation [63] causes oxidative stress, which disrupts the repair processes in the wound and has an adverse effect on the entire body, which translates into, for example, increased insulin resistance and, consequently, increased glycemia. This triggers a vicious cycle mechanism as normalization of glycemia is necessary for wound healing. The changes taking place during local infection and their consequences are comprehensively described by Huang et al. [64]. In a hyperglycemic environment, the activity of antioxidant enzymes is reduced, the number of advanced glycation end products (AGEs) increases, which enhances the hypoxia of local tissues, which is further intensified by oxygen-consuming pathogens (in the case of infection) and inflammatory cells [64]. There is data supporting the impact of exercise on the reduction of the severity of local inflammation, regardless of other standard actions taken in this regard [30,65,66,67,68,69]. This is due to the mechanisms activated during exercise, which limit the production of pro-inflammatory cytokines and increase the secretion of anti-inflammatory myokines. The latter also enhances local angiogenesis [70,71], which is essential in the treatment of ischaemic wounds. Lowering excessive blood glucose levels is important at every stage of wound healing, as high glycemia inhibits the proliferation and differentiation of fibroblasts, impairs the secretion of transforming growth factor-β (TGFβ), and reduces collagen synthesis [71].

Additionally, in such a hyperglycemic environment, the activity of endothelial nitric oxide synthase (eNOS) is inhibited, which leads to vascular damage [72]. Hyperglycemia and vascular damage are thus closely related. Moreover, a lack of muscle contraction and joint flexion (natural activities when walking) can cause swelling, which further impedes microcirculatory flow and stiffens joints, reducing their range of motion. Various forms of planned activity, such as Buerger exercises [73,74,75], active range of motion exercises, or stretching [52], contribute to improving local circulation, which creates better conditions for the healing of changes in the foot and prevent muscle atrophy.

Furthermore, improving the ROM is important for the patient’s future [76,77] as its limitation again causes abnormal load on the plantar surface of the foot, callus formation, and an incorrect operation of the muscle pump. However, some of the forms of offloading, such as TCC, may be an obstacle to performing exercises [60]. The necessity of conducting research on an offloading device that does not hinder the involvement of the affected limb in simple exercises was emphasized by the authors of one of the last articles [78].

There is considerable controversy when it comes to exercises during the active phase (open wound or inflammation) [79]. This is understandable considering the fact that most clinical trials are quite old, involve a relatively small number of patients, and are not uniform in their design [55]. This makes them difficult to interpret in terms of patient benefits. Based on the available studies, the existing knowledge is sufficient to consider properly conducted rehabilitation as necessary, even in the acute phase, with the principle of relieving the affected area of the foot and the absence of signs of general infection [80]. Consistent with the above are the aforementioned recently published research results [43], which showed significant improvement in the healing of foot ulcerations in patients with diabetes.

### 4.3. Impact of Exercises on the General Condition of a Patient with DFD

The benefits of exercise in patients with DFD are both systemic and localized [81,82,83]. Current knowledge about the mechanisms through which various forms of PA regulate different pathological processes or support physiological functions is extensive, and many of these mechanisms are well understood. A detailed summary of this knowledge, including the most recent findings, can be found in the article published by Peng et al. [84], as a detailed discussion of the effects of various forms of PA on patients is beyond the scope of this article.

Overall, the studies confirm that PA in individuals with DM exerts beneficial effects on the cardiovascular and respiratory system [85,86], microcirculation [87,88], or kidneys [89], which is not without significance considering the fact that the correlation between chronic complications and cardiovascular risk has been known for years. Improvements in insulin sensitivity, the patient’s functional and psychological state, and quality of life are also the effects of targeted activity [90].

A key therapeutic principle is the structured implementation of PA, as offloading the foot—provided there are no systemic signs of inflammation—does not necessitate full immobilization of the patient in a bed or a chair.

To perform a large part of daily activities, people involve the lower limbs. When this is limited by a disease, it should be considered whether lower limb muscle activity can be replaced with exercises that exclude the affected area of the foot or include work with the upper limbs or trunk. This is supported by the results of studies that emphasize the role of aerobic exercise in wound healing both in animals [91,92,93,94] and in humans [95], most likely through benefits for the respiratory and cardiovascular systems and improved glucose burning [42,85]. Strength training to prevent atrophy is also helpful in keeping the patient’s muscles in better shape and building muscle mass, which also translates into better glucose utilization and improves muscle tone and coordination of movements [57]. Strengthening the muscles and the patient’s general condition is also of tremendous importance if the treatment of the diabetic foot fails. If amputation and subsequent prosthetics are necessary, the patient’s energy expenditure (EE) when moving is much higher than an able-bodied person [96]. People whose condition indicates a high risk of amputation should be adequately prepared physically so that they can fully use the prosthesis in the future.

### 4.4. Discussion of Example Exercises for Patients with Diabetic Foot Disease and Contraindications or Precautions for Their Performance

Patients can exercise while sitting or lying down, depending on their overall health and the location of the ulcer. Ankle mobility exercises, such as plantar and dorsal flexion (including toes, separately), pronation and inversion or foot circles are safe and easy to perform, even for elderly people. Their role is to maintain or improve blood flow and joint mobility, they have an antithrombotic and anti-swelling effect. There are also no contraindications to performing simple exercises involving the larger joints of the lower limb (such as alternating flexion/extension at the knee or hip joint or the so-called bicycle exercise while lying on the back). In patients with heel ulcers, a prerequisite during such exercises is the ability to lift the heel during movement, for example, when bending at the knee joint, so that the patient does not rub the affected area on the ground.

Patients with an ischemic foot may also (if they tolerate pain or take pain medication) raise their lower limb and perform the exercises in that position. This is an element of the Buerger [73,74,75] exercises recommended for leg ischemia.

In the absence of contraindications to loading (e.g., when the wound involves the dorsum of the foot) a standing or at least sitting position is recommended. Energy expenditure is approximately 10% higher in the standing position compared to when a person is sitting or lying [97], therefore the change in the patient’s position should be considered beneficial for the degree of glucose utilization.

Exercises can also be performed with the involvement of the upper extremities. Strengthening the muscles of the rim and the free upper limb, apart from its overall impact on the patient’s well-being, respiratory and circulatory systems, is useful due to the patient’s potential need to move in a wheelchair or on crutches if the treatment fails and amputation is necessary [98]. The work of the arms seems to be as burdensome for the body as the work performed using the lower limbs [99], with even better synchronization of breathing with the movements of the upper limbs (vs lower limbs) during submaximal exercises [100]. However, whenever possible, patients should be encouraged to perform dynamic exercises using both the upper and lower extremities (e.g., non-weight bearing combined arm + leg cycling), if only due to the accompanying higher energy expenditure and effect on aerobic conditioning necessary with other limited types of activities [101].

Isometric exercises (alternate tensing and relaxing of muscle parts, without changing the distance of their attachments) are also a frequently used form of rehabilitation when one is unable to move or as a supplement to dynamic exercises. Due to the lack of need for additional movements, they can be performed by people with low physical fitness and poor physical condition, thus easily improving muscle strength and blood flow. Tensing muscles should be avoided in the area where sutures are placed (e.g., to bring the edges of a wound closer or minor amputation) or in the area of fractures (Charcot’s foot) so as not to cause the bones to shift relative to each other due to the tensing of the muscles. The undoubted advantage of the exercises is that they can also be performed in situations of permanent immobilization, with dressings often used for the purpose of offloading.

Learning to walk using crutches remains an element of rehabilitation, both in terms of utility (a way to move around without burdening the affected foot) and EE. Walking on crutches on flat ground or up the stairs is an effort of about 4.5 and 5.0 metabolic equivalents (METs), respectively [102]. Therefore, patients should not be persuaded to use electric wheelchairs for no reason, especially to cover short distances.

Calisthenics strengthens muscles and leads to an additional EE and good heart rate (HR) response [103]. Simple muscle-strengthening exercises, like bedside squats (only when the wound is on the upper part of the foot) [104], sit-ups, or other forms of muscle strengthening using simple instruments, such as stretching rubber bands, squeezing rubber balls, and upper limb exercises with weights, are useful in working with patients. Their role often seems small and is underestimated but the intensity of chair-assisted exercises, for example, could be at least moderate, and these exercises can give about 61% of VO2max, 67% HRmax, and about 3.9 METs loading [104].

In addition to the above-mentioned exercises, patients with foot deformity without ulceration can perform low-intensity aerobic exercises in standing or walking position after fitting an orthotic, if necessary, or with the use of made-to-measure shoes or individual orthotics. If such exercises are performed on a bicycle, the orthosis must be adapted to the foot deformity, and the need to redistribute the pressure of the sole on the pedals should also be taken into account [57]. A good exercise environment for people with deformities without ulceration is water [57].

Table 2 presents examples of simple exercises that can be performed by patients with DFD. The proposed exercises are original and reflect the authors’ clinical approach to rehabilitation in DFD.

Analyzing the recent reports, it is worth mentioning the possibilities offered by virtual therapy. The impact of some forms of virtual training on aerobic capacity, functional lower-limb strength, reaction time, and pain reduction have been documented [105]. This new and still experimental form of rehabilitation takes advantage of the well-known physiological phenomenon of activating the neural networks of the cerebral cortex and cerebral subcortex, which contain the so-called mirror neurons. After watching an exercise being performed (the element of virtual therapy), its performance is simpler and more correct due to the enhancing adaptive neuroplasticity mechanisms [106]. Although there are no studies for patients with limited mobility due to the diabetic foot, various research using mirror neuron-based therapy have proved its effectiveness on several motor functions in populations with neurological diseases [107,108,109,110], and it seems reasonable to preliminarily extrapolate the obtained results to patients with DM and foot problems until that population is studied. The introduction of virtual reality-based therapy can be an additional tool in the effective use of rehabilitation time in the aforementioned group of people, without the risk of tissue damage or other side effects.

After the ulcer has healed and the inflammation has subsided, the patient may continue the exercises performed during the acute phase of the complication. At this stage, it is also possible to introduce additional exercises with full weight-bearing activities. However, it should be taken into account that the previously affected area remains vulnerable to re-injury. For this reason, patients should protect their feet from mechanical, biological, and thermal factors by special care and by using appropriate insoles or individually fitted footwear. During the recovery phase, it is also advisable to consider vascular procedures—if they could not be performed earlier—as well as orthopedic interventions to correct foot deformities, where possible.

## 5. Conclusions

Given the absence of evidence indicating harm from properly performed exercises in patients with diabetic foot disease, and considering their documented systemic and, in some cases, local benefits, physical activity should be incorporated into DFD management. Importantly, exercises should not be limited to the affected limb. Activating other body regions can counteract the negative systemic effects of immobilization. Exercise types may include aerobic, anaerobic, isometric, calisthenics, stretching, and active joint mobility exercises, selected based on the location and severity of the foot lesions, offloading method, pain presence, and the patient’s general condition. A clearly structured, individualized exercise plan should be developed at the beginning of treatment and adjusted as the patient’s condition evolves.

The decision to include exercises and reassurance of their safety by medical staff strongly influences patient engagement. Effective rehabilitation requires an interdisciplinary approach and structured patient education to improve compliance and outcomes while minimizing complications. Figure 1 illustrates the need to balance local offloading with systemic and targeted PA—both essential to optimal DFD therapy.

Although high-quality RCTs on exercise interventions in DFD remain limited—mainly due to methodological challenges—small, well-designed interventional studies offer valuable insights and can support future meta-analyses.

## Figures and Tables

**Figure 1 biomedicines-13-01752-f001:**
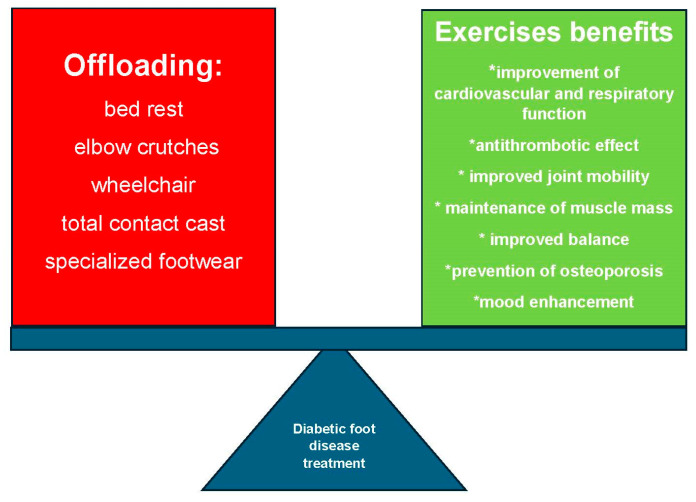
Balance between Offloading and Exercise in Diabetic Foot Disease (DFD).

**Table 2 biomedicines-13-01752-t002:** Type and examples of exercises that may benefit patients with diabetic foot disease ^A^.

Type of Exercise	Benefits	Example of Exercise
Active range of motion exercises	↓ local inflammation ↑ local circulation ↓ swelling ↑ muscle pump antithrombotic effect ↑ joints mobility improvement of plantar load ↓ callus formation ↑ flexibility	-plantar/dorsal flexion -pronation/inversion of the foot -foot circles -alternating flexion/extension of the knee joint -alternating flexion/extension of the hip joint
Stretching	↑ local circulation ↓ swelling ↑ flexibility	-attempt to touch the toes of an extended leg with your hand (Hamstring Stretch) -attempt to touch the toes of an extended leg with your hand while sitting with legs spread apart (Inner Thigh Stretch) -briefly sitting on the heels (Shin Stretch)
Aerobic	improve respiratory endurance improve circulatory endurance ↓ glucose ↑ energy expenditure	-exercises in standing or walking position after fitting an orthotic if not contraindicated -non-weight bearing combined arm + leg cycling -non-weight bearing arm ergometer
Strength	↓ risk of osteoporosis ↑ energy expenditure ↑ muscle tone ↑ muscle strength ↓ glucose (in the long-term context) ↑ blood flow ↑ flexibility improvement of coordination	-weight bearing combined arm and leg (if not contraindicated) cycling -upper and lower limbs exercises with weights -stretching rubber bands (exercise for the upper and lower extremity)
Isometric	↑ muscle strength ↑ blood flow ↓ swelling antithrombotic effect	-alternating tensing and relaxing the muscle of the calf, thigh, buttock
Other specific exercise	↑ muscle strength	-calisthenics (using natural body movements)
	↑ energy expenditure	-calisthenics, Buerger exercises
	↑ local circulation	-Buerger exercises
	↑ blood flow improve respiratory endurance	-Respiratory exercises: -diaphragmatic breathing -pursed-lip breathing -resisted breathing

^A^ Sources: Compiled by the authors based on data from [6,17,18,19,21,22,23,24,25,27,28,37,38,39,41,42,43,55,57,61,63,65,68,73,74,75,77,80,90].

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
