# Peer review of "On the Merits of Targeted and Individualized Physical Exercise in Persons with Diabetic Foot Disease—From Controversies to Consensus"

_biomedicines, 2025, doi:10.3390/biomedicines13071752_

Round 1
Reviewer 1 Report
Comments and Suggestions for Authors
This review addresses a clinically significant and timely topic—the role of targeted and individualized physical exercise in patients with diabetic foot disease (DFD). However, despite the value of the subject matter, the article’s current structure and presentation diminish its potential impact and clarity.
The manuscript currently lacks a cohesive and guiding framework. The absence of clearly defined segments results in repetition of content and a fragmented presentation of ideas. This weakens the narrative coherence and hinders reader comprehension, especially for clinicians or researchers seeking to apply the insights in practice.
Given the nature of the article as a narrative or conceptual review, the inclusion of a dedicated "Methodology" section appears unnecessary and contributes little. A concise explanation within the introduction would be more appropriate.
To enhance clarity and thematic coherence, I recommend organizing the manuscript into the following structured segments:
Segment 1: Functional and Physical Impact of Diabetic Foot Disease
- Provide a chronological description of the pathophysiological cascade following the development of DFD, illustrating how complications evolve over time.
- Emphasize the distinct clinical phases of the disease:
- Prevention Phase
- Active Phase (with ulcers or inflammation)
- Recovery Phase
- Address how DFD compromises mobility, postural control, musculoskeletal integrity, and independence.
Segment 2: Local Effects of Physical Exercise in DFD
- Examine how exercise interventions impact specific anatomical structures, including:
- Muscles (atrophy, strength)
- Tendons and ligaments (flexibility, loading)
- Joints and bones (mobility, osteoarticular health)
- Align the discussion with the clinical phases outlined above to reinforce comprehension.
Segment 3: Systemic Effects of Physical Exercise in DFD
- Detail the multisystemic benefits of exercise, categorized as:
- Cardiovascular and microcirculatory improvements
- Metabolic regulation, particularly glycemic control and insulin sensitivity
- Respiratory and functional capacity
- Cognitive and psychological well-being
- Link these effects to long-term outcomes such as quality of life and risk reduction for amputation or relapse.
Segment 4: Role and Application of Different Exercise Modalities
- Building upon the information from segments 3,4, the benefits of exercise throughout the different phases could be more effectively presented by grouping them according to exercise type, for example:"
- Aerobic training
- Resistance/strength exercises
- Balance and proprioception training
- Flexibility/mobility routines
- Table 1 should be refined. Specifically:
- Replace “Range of motion” with “Active mobility exercises” or “Active range of motion exercises” to reflect standard terminology.
- Incorporate supporting references into the table to enhance scientific credibility and utility.
Segment 5: Conclusions and Practical Implications
- Summarize the key clinical messages for rehabilitation professionals.
- Highlight the need for individualized protocols, interdisciplinary coordination, and patient education.
- Suggest future directions for research, particularly randomized controlled trials with phase-specific interventions.
Suggested Enhancements
- The inclusion of schematic diagrams, clinical flowcharts, or conceptual models would greatly enhance the reader’s understanding and break the monotony of dense textual content.
- For example:
- A timeline of disease progression and exercise intervention windows
- A matrix linking exercise types to therapeutic goals by disease phase
While the article addresses a clinically underexplored but increasingly relevant issue, its current form lacks the organizational clarity and narrative flow necessary for full impact. By adopting the proposed structure and refining terminology and visual elements, the manuscript can be significantly improved to better serve both academic and clinical audiences.
Comments on the Quality of English LanguageAlthough English is not my native language, I noticed a few errors
Author Response
Dear Reviewers
Thank you very much for the exceptionally detailed analysis of our article and the numerous comments. Although we agree with most of them, it is not always possible to fully implement all the suggestions in the article. We address these details below, responding step by step to the reviewers’ remarks.
In the article text, the changes have been highlighted in yellow to facilitate their identification. This also applies to relocated sections.
Our responses below are written in bold font.
Reviewer no 1.
This review addresses a clinically significant and timely topic—the role of targeted and individualized physical exercise in patients with diabetic foot disease (DFD). However, despite the value of the subject matter, the article’s current structure and presentation diminish its potential impact and clarity.
The manuscript currently lacks a cohesive and guiding framework. The absence of clearly defined segments results in repetition of content and a fragmented presentation of ideas. This weakens the narrative coherence and hinders reader comprehension, especially for clinicians or researchers seeking to apply the insights in practice.
- Given the nature of the article as a narrative or conceptual review, the inclusion of a dedicated "Methodology" section appears unnecessary and contributes little. A concise explanation within the introduction would be more appropriate.
The authors entered the information according to the provided template. We fully agree that it is not appropriate for this type of article. Therefore, we sincerely thank you for this remark; the titles of chapters and subchapters have been changed.
- To enhance clarity and thematic coherence, I recommend organizing the manuscript into the following structured segments:
As the proposed structure of the article’s subsections was accepted by the other Reviewers, we have made only necessary adjustments to accommodate all comments. Nevertheless, we made sure to avoid repetition of information, as you noted. We have modified some subsection titles and introduced new ones. A detailed discussion of the various components of physical activity and their effects, broken down by type of exercise, was not feasible due to word limit constraints. However, we aimed to maintain logical transitions between sections and retained the table to illustrate how different types of exercises can improve pathological conditions associated with DFD.
2.1 Segment 1: Functional and Physical Impact of Diabetic Foot Disease
Provide a chronological description of the pathophysiological cascade following the development of DFD, illustrating how complications evolve over time.
Emphasize the distinct clinical phases of the disease:
Prevention Phase
Active Phase (with ulcers or inflammation)
Recovery Phase
Due to space limitations, the article does not include a detailed discussion of the pathophysiological cascade leading to DFD. Instead, we proposed a table that allows the reader to follow the key mechanisms and consequences, as you suggested." Table 1: Functional and physical consequences of diabetic foot disease.”
The prevention phase was discussed in a few sentences, referring to both neuropathy and blood supply: „There is ample evidence (…)and blood supply to the limbs is assessed.” This fragment was distinguished with a separate paragraph. Following it, we smoothly transition—while discussing the causes—into the active phase, which is the main focus of the article : „ In some cases (…) is the occurrence of diabetic foot disease (DFD).”
The article focuses exclusively on the use of exercise during the active phase of the disease—not during the prevention phase or the recovery phase. These two phases, which were omitted, are only briefly mentioned. It was not the authors’ intention to discuss them in detail, nor does the article’s length allow for such an expansion. We chose to address patient activity during the acute phase of DFD due to the scarcity of existing literature on this topic and the particularly difficult decisions therapists face at this stage, given the often contradictory recommendations.
The entire article is devoted to the active phase of DFD. However, we emphasize, in the first text, the role of exercise not only during the acute phase e.g.: „Additionally, improving the ROM is important for the patient's future as its limitation again causes abnormal load on the plantar surface of the foot, callus formation and an incorrect operation of the muscle pump „ or „Strengthening the muscles and the patient's general condition is of tremendous importance also if the treatment of the diabetic foot fails. If amputation and subsequent prosthetics are necessary, the patient's energy expenditure (EE) when moving is much higher than in able-bodied persons. People whose condition indicates a high risk of amputation should be adequately prepared physically so that they can fully use the prosthesis in the future.”
Despite of these, we have added based on your sugestions: „After the ulcer has healed and the inflammation has subsided, the patient may continue the exercises performed during the acute phase of the complication. At this stage, it is also possible to introduce additional exercises with full weight-bearing activities. However, it should be taken into account that the previously affected area remains vulnerable to re-injury. For this reason, patients should protect their feet from mechanical, biological, and thermal factors by special care and using appropriate insoles or individually fitted footwear. During the recovery phase, it is also advisable to consider vascular procedures—if they could not be performed earlier—as well as orthopedic interventions to correct foot deformities, where possible.”
Address how DFD compromises mobility, postural control, musculoskeletal integrity, and independence.
What you are requesting is so extensive that it would cause the article to be rejected due to its length and the core idea. We have added a few information on the topic you requested (mainly in „Consequences of diabetic foot disease” section with Table 1). A reader seeking a more detailed explanation can supplement their knowledge by studying the literature references we have provided.
2.2 Segment 2: Local Effects of Physical Exercise in DFD
Examine how exercise interventions impact specific anatomical structures, including:
Muscles (atrophy, strength)
Tendons and ligaments (flexibility, loading)
Joints and bones (mobility, osteoarticular health)
Align the discussion with the clinical phases outlined above to reinforce comprehension.
The section on local and systemic benefits of exercise aims to highlight the most important advantages that patients may gain if appropriate physical activity is incorporated into treatment. Examples of such exercises, the benefits they offer, and simplified implementation guidelines are presented in Table 2. We do not comment clinical phases of the DFD ( explanation is above).
2.3 Segment 3: Systemic Effects of Physical Exercise in DFD
Detail the multisystemic benefits of exercise, categorized as:
Cardiovascular and microcirculatory improvements
Metabolic regulation, particularly glycemic control and insulin sensitivity
Respiratory and functional capacity
Cognitive and psychological well-being
Link these effects to long-term outcomes such as quality of life and risk reduction for amputation or relapse.
We sincerely apologize, but a detailed discussion of the systemic effects of exercise on the patient’s overall condition is beyond the scope of this article. This topic is typically addressed in entire textbook chapters. Even the impact of physical activity on glycemic control alone would require a separate publication, despite the fact that in patients with DFD, the scope of recommended exercises is understandably limited. Nevertheless, both in the Introduction and Table 2, we have provided a general overview of the benefits of exercise for the human body.
- Segment 4: Role and Application of Different Exercise Modalities
Building upon the information from segments 3,4, the benefits of exercise throughout the different phases could be more effectively presented by grouping them according to exercise type, for example:"
Aerobic training
Resistance/strength exercises
Balance and proprioception training
Flexibility/mobility routines
The article focuses exclusively on physical exercises during the acute phase of DFD, and this has been the central focus throughout the manuscript. Table 2 provides concise information on the effects of various types of exercises on the local or general condition of the patient, along with examples. Chapter 4 is also entirely dedicated to this discussion. More detailed discussion of the effects of these exercises goes beyond the scope of this article also because of its lengh.
- Table 1 should be refined. Specifically:
Replace “Range of motion” with “Active mobility exercises” or “Active range of motion exercises” to reflect standard terminology.
Incorporate supporting references into the table to enhance scientific credibility and utility.
All suggested changes have been incorporated into the manuscript.
- Segment 5: Conclusions and Practical Implications
Summarize the key clinical messages for rehabilitation professionals.
Highlight the need for individualized protocols, interdisciplinary coordination, and patient education.
Suggest future directions for research, particularly randomized controlled trials with phase-specific interventions.
Suggested Enhancements
We have completely revised the Conclusions section to avoid lengthening the article and repeating information from the main text, while at the same time addressing your suggestion.
- The inclusion of schematic diagrams, clinical flowcharts, or conceptual models would greatly enhance the reader’s understanding and break the monotony of dense textual content.
For example:
A timeline of disease progression and exercise intervention windows
A matrix linking exercise types to therapeutic goals by disease phase
Therapeutic windows depend on both the general and local condition of the patient, as highlighted in the text. They are highly individual due to the nature of the complication and its consequences.
We do not discuss the phases of the disease. The therapeutic goals of the acute phase are included in the main text e.g. „ (…) to maintain or improve blood flow and joint mobility…” and many other examples are provided throughout the text.
We have included an additional table and an illustration highlighting the need to maintain a balance between offloading and physical exercise in order to enrich the message and improve content clarity.
As university employees, we are required to submit every manuscript for language editing, thus we have certificate for the text. Our institution has a contract with a professional editing company that performs this service for a fee. Of course, we understand that, despite this process, some linguistic errors may remain in the text. For this reason, we have also subjected the manuscript to AI-based language correction. Nevertheless, we kindly ask you not to hesitate to indicate any specific sections that appear unclear or linguistically incorrect. This will allow us to file a formal complaint with the editing provider.
We hope that the current version of the manuscript will be satisfactory to both the Reviewers and the Editor.
Kind regards
Corresponding author
Reviewer 2 Report
Comments and Suggestions for Authors
Authors present a work addressing: ‘On the merits of targeted and individualized physical exercise in persons with diabetic foot disease – from controversies to consensus’. The aim of the study was indicates to the role that exercise, when performed with individualized precautions, can play in improving the condition of a patient with a wound and/or foot deformity due to diabetic foot disease. The general conclusion demonstrates that harmful effects of properly performed exercises in patients with diabetic foot disease and taking into account the documented benefits of exercises on the general condition of the patient, and in some cases the local condition, it should be recommended that their performance should be part of the diabetic foot disease therapy
I have following comments:
General comments:
I strongly recommend that the manuscript be thoroughly revised by a professional native speaker with experience in scientific or medical writing, or that the authors consider using a certified editing service. Improving the linguistic quality will greatly enhance the manuscript’s readability, credibility, and overall impact.
Major
1. The innovative character of the paper may be questionable, given that only 28 out of 97 references (29 %) were published within the last five years, whereas 38 references (39 %) date back to before 2015. The authors should incorporate more recent literature, which could strengthen the rationale and highlight the relevance of the study more clearly.
2. The introduction begins with a very general statement on exercise as a component of diabetes treatment, which does not sufficiently reflect the specific focus of the article—namely, targeted and individualized exercise in the context of diabetic foot disease. For a review article with this title, the opening paragraph should more clearly introduce the clinical relevance and complexity of diabetic foot pathology and the rationale for discussing individualized physical activity. A more focused and topic-relevant introduction would better engage the reader and set an appropriate scientific tone. Moreover, the introduction lacks epidemiological data concerning the prevalence and burden of diabetic foot disease. Including statistics on incidence, recurrence, amputation rates, and related healthcare costs would provide important context for the topic and reinforce the need for individualized exercise strategies in this patient population.
3. Throughout the manuscript, it remains unclear whether the discussion refers to type 1 diabetes (T1DM), type 2 diabetes (T2DM), or both. This distinction is critical, as the pathophysiology, prevalence, and response to physical activity differ substantially between the two. For instance, the reference to increased insulin resistance as a consequence of physical inactivity strongly suggests a focus on T2DM (line 46), but this is not explicitly stated. I recommend that the authors clarify the target population and specify whether their recommendations apply to individuals with type 2 diabetes, type 1 diabetes, or both, and provide justification accordingly.
4. In line 27, the authors use the abbreviation DM for “diabetes,” but it is not specified that this refers to diabetes mellitus. Since “diabetes” may refer to other conditions (e.g., diabetes insipidus), it is recommended to clearly state “diabetes mellitus (DM)” at first mention for clarity and precision.
5. The number of references [12–20] cited to support the statement regarding the reduction of diabetes-related foot pathologies seems excessive. It may be more appropriate to select a few representative or higher-quality studies to improve clarity and avoid over-referencing. Among the nine references cited to support the statement on diabetes-related foot pathologies ([12–20]), only one was published within the last five years. Considering the rapid evolution of clinical evidence in this area, it would be advisable to update the references and include more recent studies, preferably systematic reviews or high-quality clinical trials published in the last 5–7 years.
6. The introduction section appears too condensed in relation to the number of references cited (34 references over approximately one A4 page). Considering that the full manuscript includes 97 references, this suggests an imbalance in literature coverage. I recommend expanding the introduction to provide a more comprehensive and structured overview of current knowledge, highlight key recent studies, and better justify the research gap and study objectives. A stronger, more context-rich introduction would greatly improve the scientific grounding of the manuscript.
7. The section titled “Results” may not be appropriate for a narrative review article, as it suggests the presentation of original data or outcomes from a study, which is not the case here. Since the content that follows is interpretive in nature and focuses on summarizing and contextualizing findings from previous research, I suggest renaming the section to “Discussion,” “Discussion and Implications,” „Evidence Synthesis and Clinical Implications”or a similar heading that more accurately reflects the structure and purpose of a review article.
8. The section 3.1. presents important clinical considerations regarding the management of diabetic foot disease (DFD), particularly the tension between the need for offloading and the benefits of physical activity. However, the structure and focus of the text could be improved. The initial part of the paragraph discusses pharmacological treatment and revascularization, which, while relevant to DFD in general, seems somewhat tangential to the main topic of the review. I suggest restructuring this part to clearly distinguish between standard DFD management and the core discussion on the implications of physical inactivity and the role of exercise. In addition, some sentences are overly dense and could be rephrased for clarity and flow. A more coherent and focused narrative would help better communicate the clinical dilemma and the need for individualized approaches in balancing offloading with functional mobility.
9. Table 1 provides a useful overview of exercise types and examples relevant to patients with diabetic foot disease. However, given the multifactorial nature of this condition and the clinical dilemmas described in the text, the manuscript would benefit from an additional figure or table. For example, a schematic diagram illustrating the balance between offloading and physical activity, or a table summarizing current clinical studies on exercise interventions in DFD, would significantly enhance the clarity, scientific value, and usability of the review.
10. The authors should clearly emphasize the unique aspects of this review, explaining why it is important and what makes their contribution valuable to the field.
Minor:
1. In Table 1, the font should be adjusted to Palatino to ensure consistency with the rest of the manuscript and formatting guidelines.
2. I recommend that the authors include a discussion of the limitations of their literature analysis to provide a balanced and transparent overview of the review’s scope and potential weaknesses.
I strongly recommend that the manuscript be thoroughly revised by a professional native speaker with experience in scientific or medical writing, or that the authors consider using a certified editing service. Improving the linguistic quality will greatly enhance the manuscript’s readability, credibility, and overall impact.
Author Response
Dear Reviewers
Thank you very much for the exceptionally detailed analysis of our article and the numerous comments. Although we agree with most of them, it is not always possible to fully implement all the suggestions in the article. We address these details below, responding step by step to the reviewers’ remarks.
In the article text, the changes have been highlighted in yellow to facilitate their identification. This also applies to relocated sections.
Our responses below are written in bold font.
Rev no 2:
Authors present a work addressing: ‘On the merits of targeted and individualized physical exercise in persons with diabetic foot disease – from controversies to consensus’. The aim of the study was indicates to the role that exercise, when performed with individualized precautions, can play in improving the condition of a patient with a wound and/or foot deformity due to diabetic foot disease. The general conclusion demonstrates that harmful effects of properly performed exercises in patients with diabetic foot disease and taking into account the documented benefits of exercises on the general condition of the patient, and in some cases the local condition, it should be recommended that their performance should be part of the diabetic foot disease therapy
I have following comments:
General comments:
I strongly recommend that the manuscript be thoroughly revised by a professional native speaker with experience in scientific or medical writing, or that the authors consider using a certified editing service. Improving the linguistic quality will greatly enhance the manuscript’s readability, credibility, and overall impact.
As university employees, we are required to submit every manuscript for language editing, thus we have certificate for the text. Our institution has a contract with a professional editing company that performs this service for a fee. Of course, we understand that, despite this process, some linguistic errors may remain in the text. For this reason, we have also subjected the manuscript to AI-based language correction. Nevertheless, we kindly ask you not to hesitate to indicate any specific sections that appear unclear or linguistically incorrect. This will allow us to file a formal complaint with the editing provider.
Major
- The innovative character of the paper may be questionable, given that only 28 out of 97 references (29 %) were published within the last five years, whereas 38 references (39 %) date back to before 2015. The authors should incorporate more recent literature, which could strengthen the rationale and highlight the relevance of the study more clearly.
There are actually not many studies directly related to the topic we address in the main text. Apart from two references (from 2022 and 2025) we have added, we did not find any others that contribute significantly to the topic under discussion.
- The introduction begins with a very general statement on exercise as a component of diabetes treatment, which does not sufficiently reflect the specific focus of the article—namely, targeted and individualized exercise in the context of diabetic foot disease. For a review article with this title, the opening paragraph should more clearly introduce the clinical relevance and complexity of diabetic foot pathology and the rationale for discussing individualized physical activity. A more focused and topic-relevant introduction would better engage the reader and set an appropriate scientific tone. Moreover, the introduction lacks epidemiological data concerning the prevalence and burden of diabetic foot disease. Including statistics on incidence, recurrence, amputation rates, and related healthcare costs would provide important context for the topic and reinforce the need for individualized exercise strategies in this patient population.
The most important information has been added. Please note the increasing length of the text and the growing number of references, which may not be accepted by the editor.
- Throughout the manuscript, it remains unclear whether the discussion refers to type 1 diabetes (T1DM), type 2 diabetes (T2DM), or both. This distinction is critical, as the pathophysiology, prevalence, and response to physical activity differ substantially between the two. For instance, the reference to increased insulin resistance as a consequence of physical inactivity strongly suggests a focus on T2DM (line 46), but this is not explicitly stated. I recommend that the authors clarify the target population and specify whether their recommendations apply to individuals with type 2 diabetes, type 1 diabetes, or both, and provide justification accordingly.
DFD, like all chronic complications of diabetes, occurs in every type of diabetes! Although insulin resistance is rightly associated with type 2 diabetes, it also appears, for example, during inflammation, complicating treatment. Thus, the occurrence of DFD also generates transient insulin resistance. The omission of specifying the type of diabetes by the authors was intentional!
The pathomechanisms and treatment of this condition do not depend on the type of diabetes either.
- In line 27, the authors use the abbreviation DM for “diabetes,” but it is not specified that this refers to diabetes mellitus. Since “diabetes” may refer to other conditions (e.g., diabetes insipidus), it is recommended to clearly state “diabetes mellitus (DM)” at first mention for clarity and precision.
We apologize for this error—it has been corrected.
- The number of references [12–20] cited to support the statement regarding the reduction of diabetes-related foot pathologies seems excessive. It may be more appropriate to select a few representative or higher-quality studies to improve clarity and avoid over-referencing. Among the nine references cited to support the statement on diabetes-related foot pathologies ([12–20]), only one was published within the last five years. Considering the rapid evolution of clinical evidence in this area, it would be advisable to update the references and include more recent studies, preferably systematic reviews or high-quality clinical trials published in the last 5–7 years.
Each of the cited references addresses a slightly different issue: basic research, clinical studies, or reviews, RCT and they relate to different forms of exercise, such as aerobic or localized exercises. This literature does not pertain to a single thesis. As mentioned in the sentence you referred to, it concerns numerous pathologies that ultimately lead to DFD, quoting : "…and reduces the risk of pathologies typical of diabetes occurring in that area." Following your suggestion, we adopted a more critical approach to this topic; however, instead of the three removed references, we added others, including an article from 2025 that was not available to us at the time of writing the paper. No other new important documents we have found.
- The introduction section appears too condensed in relation to the number of references cited (34 references over approximately one A4 page). Considering that the full manuscript includes 97 references, this suggests an imbalance in literature coverage. I recommend expanding the introduction to provide a more comprehensive and structured overview of current knowledge, highlight key recent studies, and better justify the research gap and study objectives. A stronger, more context-rich introduction would greatly improve the scientific grounding of the manuscript.
There are actually not many studies directly related to the topic we address in the main text. However, the introduction covers information both on DFD and on physical activity in general (including recommendations, basic research, RCTs, and other clinical studies; definitions). Because these topics are combined (diabetes, diabetic foot, offloading, exercise), each requiring at least a brief discussion, the number of citations is disproportionately large. Additionally, at the Reviewer’s request, we had to further expand the introduction, which only increased this disproportion. We believe this is unavoidable.
Please also find how we explain the necessary to summarize this topic: „However, even in this document, the lack of precise recommendations regarding which exercises should be prescribed and what their potential benefits are is emphasized. Similarly, in the IWGDF guidelines - the most authoritative document in this field- 29 recommendations are provided for managing DFD, but none of them refer to exercise. This situation results from the limited number of RCTs (Randomized Controlled Trials) dedicated to this topic that have included a sufficiently large number of patients in the intervention.”
- The section titled “Results” may not be appropriate for a narrative review article, as it suggests the presentation of original data or outcomes from a study, which is not the case here. Since the content that follows is interpretive in nature and focuses on summarizing and contextualizing findings from previous research, I suggest renaming the section to “Discussion,” “Discussion and Implications,” „Evidence Synthesis and Clinical Implications”or a similar heading that more accurately reflects the structure and purpose of a review article.
The authors entered the information according to the provided template. We fully agree that it is not appropriate for this type of article. Therefore, we sincerely thank you for this remark; the titles of chapters and subchapters have been changed.
- The section 3.1. presents important clinical considerations regarding the management of diabetic foot disease (DFD), particularly the tension between the need for offloading and the benefits of physical activity. However, the structure and focus of the text could be improved. The initial part of the paragraph discusses pharmacological treatment and revascularization, which, while relevant to DFD in general, seems somewhat tangential to the main topic of the review. I suggest restructuring this part to clearly distinguish between standard DFD management and the core discussion on the implications of physical inactivity and the role of exercise. In addition, some sentences are overly dense and could be rephrased for clarity and flow. A more coherent and focused narrative would help better communicate the clinical dilemma and the need for individualized approaches in balancing offloading with functional mobility.
Some sentences have been simplified, and a smoother transition has been added between the information about actions other than adding exercises.
- Table 1 provides a useful overview of exercise types and examples relevant to patients with diabetic foot disease. However, given the multifactorial nature of this condition and the clinical dilemmas described in the text, the manuscript would benefit from an additional figure or table. For example, a schematic diagram illustrating the balance between offloading and physical activity, or a table summarizing current clinical studies on exercise interventions in DFD, would significantly enhance the clarity, scientific value, and usability of the review.
A graphical summary illustrating the balance between exercise and offloading has been added.
- The authors should clearly emphasize the unique aspects of this review, explaining why it is important and what makes their contribution valuable to the field.
The information has been added to the Conclusions according to your suggestion.
Minor:
- In Table 1, the font should be adjusted to Palatino to ensure consistency with the rest of the manuscript and formatting guidelines.
Thank you for this remark. The font has been changed.
- I recommend that the authors include a discussion of the limitations of their literature analysis to provide a balanced and transparent overview of the review’s scope and potential weaknesses.
We have added information regarding the potential adverse effects of exercise in DFD treatment, as well as a reference to a publication that analyzes this issue.
We hope that the current version of the manuscript will be satisfactory to both the Reviewers and the Editor.
Kind regrads
Corresponding author
Reviewer 3 Report
Comments and Suggestions for Authors
On the merits of targeted and individualized physical exercise in persons with diabetic foot disease – from controversies to consensus.
The review aims to consolidate the knowledge, highlighting the importance of physical activity in DFD. The approach and the overall concept of the review are good. However, authors should add more detailed sections to make the review more impactful.
- A section should be added describing the hyperglycemia-induced progression of DFD and its pathological physiological mechanism. A schematic figure should also be included in this section to make the concept more interesting.
- Section 3.3 describing the impact of exercise on the general conditions of DFD needs to be expanded further by discussing each condition in detail.
- A schematic representation should be added in the conclusion depicting the possible mechanism of the effect of exercise in DFD in light of the present review.
Author Response
Dear Reviewers
Thank you very much for the exceptionally detailed analysis of our article and the numerous comments. Although we agree with most of them, it is not always possible to fully implement all the suggestions in the article. We address these details below, responding step by step to the reviewers’ remarks.
In the article text, the changes have been highlighted in yellow to facilitate their identification. This also applies to relocated sections.
Our responses below are written in bold font.
Rev no 3
The review aims to consolidate the knowledge, highlighting the importance of physical activity in DFD. The approach and the overall concept of the review are good. However, authors should add more detailed sections to make the review more impactful.
- A section should be added describing the hyperglycemia-induced progression of DFD and its pathological physiological mechanism. A schematic figure should also be included in this section to make the concept more interesting.
We sincerely apologize, but the article focuses on a very specific clinical issue—namely, the indications for including physical exercise in the treatment process of patients with diabetic foot disease (DFD). A detailed discussion of the pathophysiological mechanisms leading to the development of DFD falls outside the scope of this review due to space limitations. These mechanisms are well known to professionals involved in the care of patients with diabetes, and the article includes references to authoritative sources that cover this information, such as the IWGDF website. The basic information can be also found in the Table 1 which was added.
- Section 3.3 describing the impact of exercise on the general conditions of DFD needs to be expanded further by discussing each condition in detail.
We understand your concern regarding the limited amount of detailed information presented in the section and in Table 2. However, the length of the article is strictly constrained. We have therefore included the most up-to-date literature on the topic, in which the mechanisms underlying the effects of physical activity on patients with diabetes—including those with complications—are discussed in detail.
- A schematic representation should be added in the conclusion depicting the possible mechanism of the effect of exercise in DFD in light of the present review
Table 2 is dedicated to this issue. A graphical summary illustrating the balance between exercise and offloading has been added.
As university employees, we are required to submit every manuscript for language editing, thus we have certificate for the text. Our institution has a contract with a professional editing company that performs this service for a fee. Of course, we understand that, despite this process, some linguistic errors may remain in the text. For this reason, we have also subjected the manuscript to AI-based language correction. Nevertheless, we kindly ask you not to hesitate to indicate any specific sections that appear unclear or linguistically incorrect. This will allow us to file a formal complaint with the editing provider.
We hope that the current version of the manuscript will be satisfactory to both the reviewers and the editor.
Kind regards
Corresponding author
Round 2
Reviewer 1 Report
Comments and Suggestions for Authors
Thank you for carefully addressing the suggested modifications and providing additional clarifications. I believe the authors have successfully reconciled the various elements involved—namely, the reviewers' suggestions, the original vision of the manuscript, and editorial requirements.
As reviewers, we often seek comprehensive explanations of topics we find particularly relevant, hoping to ensure all pertinent information is shared. However, we sometimes overlook other practical constraints.
In this case, the authors have made judicious and necessary revisions that significantly enhance the article's impact. I therefore have no objection to its publication. That said, I recommend minor textual adjustments:
- Lines 45, 78, 379: Replace "Physical activity" with "PA" for consistency.
- Lines 390–396: The text currently states "Figure 1 illustrates..."—this appears to be a typographical error, as no figures are included in the manuscript.
Reviewer 2 Report
Comments and Suggestions for Authors
The authors responded adequately to my comments. I therefore recommend the article for publication as it stands.
Reviewer 3 Report
Comments and Suggestions for Authors
The author's reply is satisfactory.